# Associations of statin adherence and lipid targets with adverse outcomes in myocardial infarction survivors: a retrospective cohort study

Rosemary Brown ![ORCID],[1] Jim Lewsey ![ORCID],[2] Sarah Wild,[3] Jennifer Logue ![ORCID],[4] Paul Welsh[1]

JL and PW are joint senior authors.

[1]Institute of Cardiovascular and Medical Sciences, University of Glasgow, Glasgow, UK
[2]Institute of Health & Wellbeing, University of Glasgow, Glasgow, UK
[3]Usher Institute, University of Edinburgh, Edinburgh, UK
[4]Lancaster Medical School, Lancaster University, Lancaster, UK

**Correspondence to**
Dr Paul Welsh;
Paul.Welsh@glasgow.ac.uk

## ABSTRACT

**Objectives** To examine associations between statin adherence and lipid target achievement in myocardial infarction (MI) survivors, and their associations with mortality and recurrent MIs.

**Design** Retrospective cohort study using linked clinical records within the National Health Service Greater Glasgow and Clyde (NHS GGC) Data Safe Haven.

**Setting** Routine clinical practice in the NHS GGC area between January 2009 and July 2017.

**Participants** Patients ≥18 years who experienced a non-fatal MI hospital admission (ICD10: I21, I22) between January 2009 and July 2014 (n=11 031), followed up from the date of MI admission until July 2017 or death, whichever occurred first.

**Primary and secondary outcome measures** Statin adherence was estimated using encashed prescriptions and lipid results from routine biochemistry data. Primary lipid and statin adherence targets were LDL ≤1.8 mmol/L and adherence ≥50%, and were related to all-cause death, deaths due to cardiovascular disease (CVD) (ICD10: I00–I99 as the underlying cause), and recurrent MI in unadjusted models and models adjusting for age, sex, socioeconomic deprivation and year of MI.

**Results** Over 4.5 years follow-up, 76% achieved LDL ≤1.8 mmol/L, and 84.5% had average adherence ≥50%. Patients with adherence <50% had an increased risk of not meeting LDL ≤1.8 mmol/L, in adjusted models (OR 2.03, 95% CI 1.78 to 2.31, p<0.0001). In univariable models, not meeting LDL ≤1.8 mmol/L was associated with increased risks of all-cause mortality (HR 1.27, 95% CI 1.16 to 1.39, p<0.0001) and CVD mortality (HR 1.29, 95% CI 1.11 to 1.51, p=0.0013). Adherence <50% was associated with increased risks of all-cause mortality (HR 1.58, 95% CI 1.44 to 1.74, p<0.0001) and CVD mortality (HR 1.60, 95% CI 1.36 to 1.88, p<0.0001). Adjustment for confounders did not abrogate these associations. Neither exposure was associated with recurrent MIs.

**Conclusions** Non-achievement of lipid and adherence targets are associated with increased risks of all-cause and CVD mortality. Further work is required to optimise their use to improve outcomes in clinical practice.

## STRENGTHS AND LIMITATIONS OF THIS STUDY

⇒ Data were derived from routine clinical practice records within a large health board in the West of Scotland, with average duration of follow-up of 4.5 years.

⇒ The use of encashed prescriptions in estimating adherence may be stronger evidence of medicines use over issued prescriptions.

⇒ In mortality analysis, time-window bias could have arisen as patients with events had reduced opportunities to achieve lipid targets or improve average statin adherence.

⇒ Suitable baseline lipid results were substantially missing within secondary prevention populations (~70%), limiting the utility of percentage change targets in this analysis.

## INTRODUCTION

Those with established cardiovascular disease (CVD) (ie, secondary prevention) are universally identified and targeted in risk management strategies due to their markedly higher rates of further cardiovascular events and overall mortality.[1 2] A cornerstone of these strategies is lipid-lowering medications, where statins are the first-line therapy.

Adherence to these medications and attaining lipid targets are key components of a patient review to address their cardiovascular risk in Scotland and the rest of the UK.[3 4] Indeed, achievement of target lipid levels (set by clinical guidelines) has been associated with a reduced likelihood of further events and mortality.[1 5] The association between adherence and the risk of cardiovascular events has also been shown in primary and secondary prevention in observational data, although adherence was measured using prescriptions issued rather than by drugs dispensed.[6 7]

There is evidence to suggest adherence is higher in secondary prevention than primary

prevention,[8] and analyses have found reasonable adherence in secondary prevention cohorts; between 68% and 75% had statin adherence ≥80% in the year following an event.[9 10] However, this was not translated into target lipid levels, where fewer than half of the patients achieved these in the same period.[10] Despite this, many of the risk factors associated with CVD have also been shown to be associated with adherence and achievement of target lipid levels. For example, men,[11 12] and, in primary prevention, those with a diagnosis of diabetes and hypertension[12] are more likely to be adherent. In secondary prevention, however, insulin use has been associated with non-adherence,[13] although in a separate secondary prevention cohort, those with a diabetes diagnosis were more likely to meet lipid targets.[10] Therefore, there superficially appears to be differences between lipid-lowering medication adherence and lipid target achievement. This leads to uncertainties in clinical guidelines and in clinical practice as to whether adherence or lipid targets are more useful surrogates of risk reduction.[14]

This analysis, using routinely collected data available for a secondary prevention population representative of a UK health board region, therefore, seeks to compare and contrast statin adherence and lipid target achievement, and compare their associations with mortality and recurrent myocardial infarctions (MIs). This analysis also investigates which is more strongly associated with outcomes, and therefore relevant for clinical decision making.

## METHODS
### Data
The data used for this analysis were a subset of an extract of all individuals in National Health Service (NHS) Greater Glasgow and Clyde (GGC), a UK NHS region in the West of Scotland, who had a lipid profile result or a prescription for a statin, ezetimibe or PCSK9 inhibitor before 29 December 2017. Data extracted included demographics, laboratory results, dispensed prescriptions, hospital admissions, death certificates and diabetes diagnoses.

To maximise completeness, patients were followed up between 1 January 2009 and 31 July 2017 (inclusive). We excluded patients who resided outside the region, as these individuals are likely to have travelled into the region for only some aspects of their care, and therefore, a complete representation of their health status was unlikely. Patients who were <18 years, died before 1 January 2009, had multiple death certificates or received apparent posthumous lipid tests or prescriptions for lipid-lowering medications more than 6 months posthumously were also excluded. Figure 1 provides an overview of the cohort derivation.

### Patient and public involvement
Neither patients nor the public were involved in the design, conduct, reporting or dissemination of this research.

### Post MI population
MIs were identified in linked hospital admission records (International Classification of Diseases, Tenth Revision (ICD-10): I21, I22 in any diagnosis field) to identify patients who experienced an MI between 1 January 2009 and 31 July 2014. These patients were followed up from their first non-fatal MI admission until 31 July 2017 or death, whichever occurred first. Patients who only experienced fatal MIs (death occurred during admission or within 30 days of discharge), whose statin adherence was >200%, or were missing a socioeconomic deprivation index were removed. Year-long time windows were constructed for each patient, starting from their baseline admission date.

### Statin adherence
Statin prescription dispensing records were used to estimate patients' adherence. Dispensed dates were used as this was the first date that the patient could have possession of the medication.

Prescription end dates were calculated as the dispensed date plus the day-coverage dispensed, and quantities were adjusted to reflect dose availability where overlaps across time windows or death occurred. The Medication Possession Ratio (MPR)[15] was calculated as (number of doses dispensed in time window/length of time window)*100 for each full-year time window, and mean-averaged. Average adherence was considered continuously and dichotomously, using cut-offs at 50%[16] and 80%.[15 17] Given that alternate day dosing is often implemented instances of poor tolerance of medication,[16] the 50% threshold was considered the primary exposure of interest.

### Lipid targets
Lipid profiles were collected as part of routine clinical practice in NHS Biochemistry laboratories. A lipid profile result consisted of five components: total cholesterol, total cholesterol:high density lipoprotein cholesterol (HDL), HDL, low density lipoprotein cholesterol (LDL) and triglycerides. If a patient had multiple tests within a time window, the mean value of the tests was calculated. Non-HDL cholesterol was calculated as total cholesterol minus HDL.

The primary lipid exposure of interest was the 2016 European Society of Cardiology (ESC) target for LDL cholesterol ≤1.8 mmol/L,[18] and the 2014 National Institute for Health and Care Excellence target for a ≥40% reduction in non-HDL cholesterol[3] was a secondary lipid exposure of interest. The latter, though not formally recommended as a target, is also referred to within current Scottish guidelines, where achievement is considered indicative of adequate adherence to statin therapy.[4] To assess the attainment of the non-HDL target, a pre-MI statin-naïve baseline was required and was identified as the last lipid profile test a patient had before their baseline admission that was more than 6 months (182 days) after the end date of any previous statin prescription. The

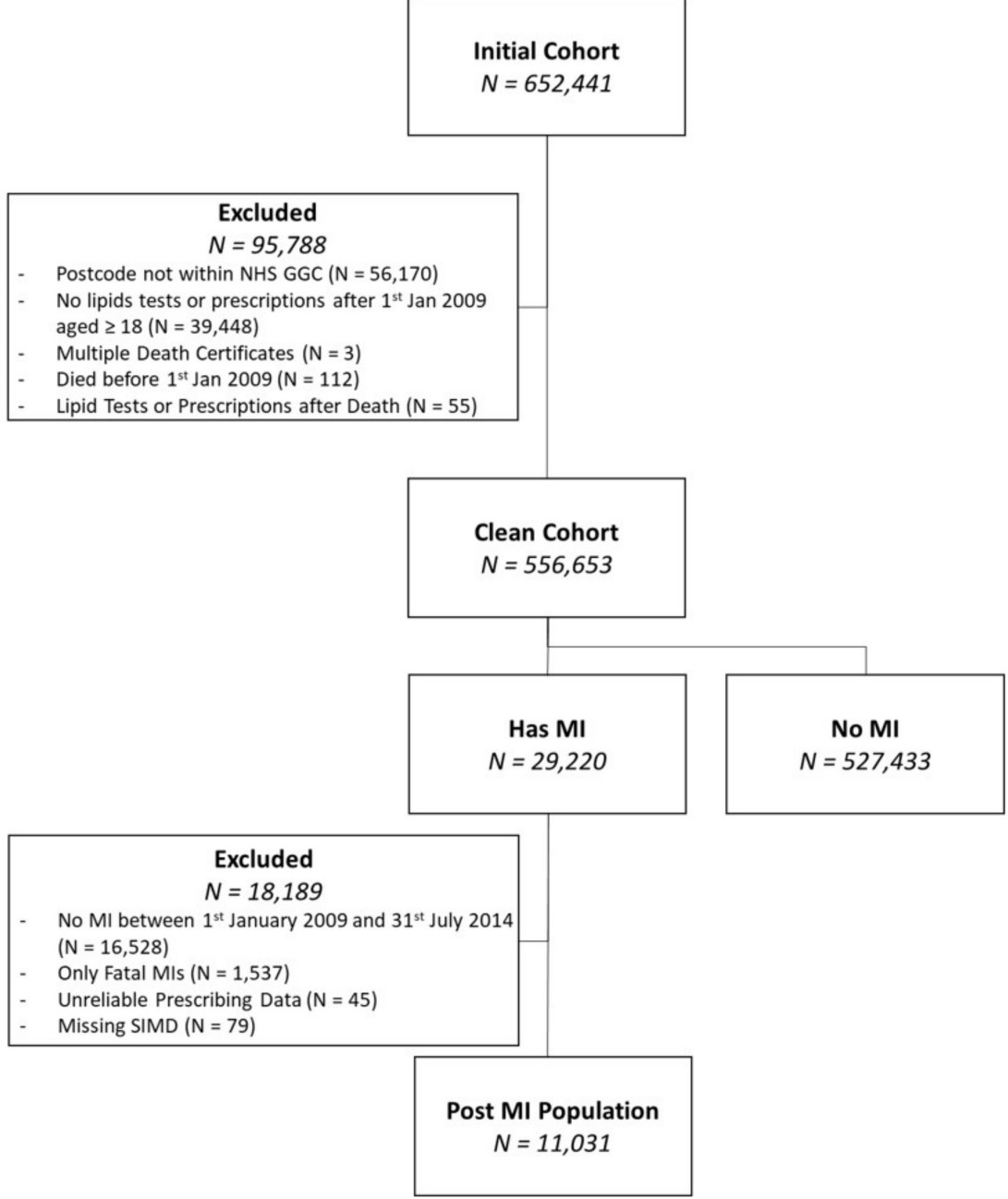

**Figure 1** Derivation of the postmyocardial infarction (MI) population. GGC, Greater Glasgow and Clyde; NHS, National Health Service; SIMD, Scottish Index of Multiple Deprivation.

need for a baseline result limited the coverage of this exposure to 30%.

### Outcomes

Underlying cause and date of death were obtained from linked death certificates and hospital discharge notices, with CVD deaths defined as ICD10: I00–I99. Patients missing a cause of death were assumed to have a non-CVD underlying cause of death. All MI events from linked hospital admission records (Scottish Morbidity Record (SMR01)) and death certificates before the end of follow-up were included as a separate outcome (ICD-10: I21, I22).

### Other covariates

Date of birth and sex was extracted from clinical records. For socioeconomic deprivation, the Scottish Index of Multiple Deprivation (SIMD) was obtained from the patients' residential postcode at baseline. Due to the higher number of the most deprived zones in this region than in Scotland in general,[19] NHS GGC-specific SIMD quintiles were derived to effectively assess the impact of deprivation within the post MI population of NHS GGC. The patient's age at MI was calculated using their date of birth and date of baseline MI. Prior MIs were defined as MI admissions occurring in the 10 years before their baseline. The date of diagnosis and the type of diabetes

was obtained from routine electronic diabetes records. The date of diagnosis was taken as the patient's first valid (ie, after the patient's date of birth) date of entry in the diabetes register. Patients were considered to have had a diagnosis of diabetes at the time of their MI if this date was before their baseline MI admission. The type of diabetes was extracted, and if multiple types were listed during a patient's history, the most common diagnosis was taken.

## Statistical analysis

All analyses were performed using R V.3.5.0,[20] and the 'forestplot',[21] 'survival'[22] and 'survminer'[23] packages. Continuous variables were summarised using mean and SD and categorical variables with the number of observations (N) and percentages. As a patient's average adherence was calculated using complete years of follow-up, only patients who survived ≥1 year were included in analyses including adherence. For analyses including lipid targets, patients were included if they reported ≥1 test result during the follow-up and were classified as meeting a target if they achieved it at any stage during the follow-up.

Due to the potential for multiple recurrent MIs to occur in an individual, and the relatively short 1-year time windows modelled, logistic regression was used to model the associations between average adherence and the achievement of lipid targets, and the association between adherence and lipid targets, with recurrent MIs. For mortality, Cox regression was conducted, and the

proportional hazards assumption was checked by visual inspection of the partial Schoenfeld residuals. For all outcomes, unadjusted and adjusted (for age at MI, sex, deprivation quintile and year of MI) models were generated, using available case analysis for each exposure. Where adherence was considered dichotomously, and for the lipid targets, achievement of the threshold or target was the reference. A sensitivity analysis for adherence was also conducted by including events that occurred in the first year of follow-up. For all analyses, a p<0.05 was considered statistically significant.

## RESULTS

### Baseline demographics

Of the 11 031 patients, 10 009 (90.7%) had an average adherence calculable, 3329 (30.2%) had ≥1 percentage change in non-HDL cholesterol result, and 9440 (85.6%) had ≥1 LDL cholesterol result. Over an average 4.5 years follow-up, the mean-average statin adherence was 79.6%, and 84.5% of patients achieved ≥50% average statin adherence and 68.9% had ≥80%. Three-quarters of patients had LDL ≤1.8 mmol/L and half achieved ≥40% reduction in non-HDL at least once during the follow-up.

Table 1 shows demographics by lipid target achievement and average adherence. Males were more likely to be adherent and achieve lipid targets. Patients who achieved ≥40% reduction in non-HDL and patients who

**Table 1** Demographics by average statin adherence (50% and 80%) and achievement of lipid targets

| Target | Average adherence ≥50% | | Average adherence ≥80% | | ≥40% reduction non-HDL* | | LDL ≤1.8 mmol/L | |
|---|---|---|---|---|---|---|---|---|
| | **Met** 8461 (84.5%) | **Not met** 1548 (15.5%) | **Met** 6894 (68.9%) | **Not met** 3115 (31.1%) | **Met** 1709 (51.3%) | **Not met** 1620 (48.7%) | **Met** 7171 (76.0%) | **Not met** 2269 (24.0%) |
| Gender | | | | | | | | |
| Male | 5337 (63.1%) | 829 (53.6%) | 4368 (63.4%) | 1798 (57.7%) | 1102 (64.5%) | 911 (56.2%) | 4566 (63.7%) | 1342 (59.1%) |
| Female | 3124 (36.9%) | 719 (46.4%) | 2526 (36.6%) | 1317 (42.3%) | 607 (35.5%) | 709 (43.8%) | 2605 (36.3%) | 927 (40.9%) |
| Age at MI (years) | | | | | | | | |
| Mean (SD) | 65.3 (13.2) | 68.1 (15.9) | 65.4 (13.0) | 66.4 (15.2) | 63.6 (12.6) | 67.0 (13.5) | 65.4 (13.2) | 65.5 (13.9) |
| SIMD 2012 quintile (NHS GGC) | | | | | | | | |
| 1 (most) | 2058 (24.3%) | 347 (22.4%) | 1652 (24.0%) | 753 (24.2%) | 398 (23.3%) | 381 (23.5%) | 1824 (25.4%) | 412 (18.2%) |
| 2 | 1979 (23.4%) | 318 (20.5%) | 1625 (23.6%) | 672 (21.6%) | 389 (22.8%) | 363 (22.4%) | 1620 (22.6%) | 551 (24.3%) |
| 3 | 1750 (20.7%) | 338 (21.8%) | 1389 (20.1%) | 699 (22.4%) | 349 (20.4%) | 349 (21.5%) | 1438 (20.1%) | 535 (23.6%) |
| 4 | 1395 (16.5%) | 289 (18.7%) | 1153 (16.7%) | 531 (17.0%) | 284 (16.6%) | 283 (17.5%) | 1140 (15.9%) | 454 (20.0%) |
| 5 (least) | 1279 (15.1%) | 256 (16.5%) | 1075 (15.6%) | 460 (14.8%) | 289 (16.9%) | 244 (15.1%) | 1149 (16.0%) | 317 (14.0%) |
| Diabetes at MI | 1535 (18.1%) | 258 (16.7%) | 1249 (18.1%) | 544 (17.5%) | 285 (16.7%) | 369 (22.8%) | 1436 (20.0%) | 312 (13.8%) |
| Type 1 | 80 (0.9%) | 26 (1.7%) | 63 (0.9%) | 43 (1.4%) | 12 (0.7%) | 29 (1.8%) | 73 (1.0%) | 32 (1.4%) |
| Type 2 | 1455 (17.2%) | 232 (15.0%) | 1186 (17.2%) | 501 (16.1%) | 273 (16.0%) | 340 (21.0%) | 1363 (19.0%) | 280 (12.3%) |
| Prior MI | 529 (6.3%) | 85 (5.5%) | 425 (6.2%) | 189 (6.1%) | 34 (2.0%) | 98 (6.0%) | 471 (6.6%) | 114 (5.0%) |
| 1 | 412 (4.9%) | 68 (4.4%) | 330 (4.8%) | 150 (4.8%) | 28 (1.6%) | 79 (4.9%) | 359 (5.0%) | 94 (4.1%) |
| >1 | 117 (1.4%) | 17 (1.1%) | 95 (1.4%) | 39 (1.3%) | 6 (0.4%) | 19 (1.2%) | 112 (1.6%) | 20 (0.9%) |

Numbers are N (%) unless otherwise specified. Percentages are calculated within columns, except for the header where percentages are calculated from total with information. For adherence, patients were only included if ≥1 complete year of follow-up was available. For lipid targets, patients were included if a necessary test result was available.
*Refers to the percentage reduction in non-HDL cholesterol from the pre-MI baseline.
MI, myocardial infarction; NHS GGC, National Health Service Greater Glasgow and Clyde; SIMD, Scottish Index of Multiple Deprivation.

were adherent (at either threshold) were younger on average. Those residing in more deprived areas were more likely to have adherence ≥50%, but this was less evident for ≥80%, and those in the most and least deprived areas were more likely to meet the LDL target. Patients with a diagnosis of diabetes were more likely to achieve LDL ≤1.8 mmol/L but were less likely to meet the non-HDL target. However, patients diagnosed with type 2 diabetes were more likely to be adherent, particularly ≥50%, while patients diagnosed with type 1 diabetes were less likely. Finally, patients with prior MIs were slightly more likely to achieve LDL ≤1.8 mmol/L and have adherence ≥50%, but the inverse was true for the non-HDL target.

### Adherence and lipid targets

Adherence was associated with lipid target achievement (online supplemental tables S1 and S2). Those with adherence <50% had an increased risk of not meeting LDL ≤1.8 mmol/L in unadjusted (OR 1.99, 95% CI 1.75 to 2.26, p<0.0001) and adjusted models (OR 2.03, 95% CI 1.78 to 2.31, p<0.0001), and had an adjusted OR of not meeting the non-HDL target of 4.5 (95% CI 3.62 to 5.54, p<0.0001). Similar patterns were observed for adherence <80%. A 10% decrease in adherence was associated with a 20% increase in odds of not meeting the non-HDL target, and a 10% increase in odds of not achieving LDL ≤1.8 mmol/L, with both increases unaltered following adjustment. In a sensitivity analysis including incomplete years of follow-up, ORs were slightly lower but retained statistical significance (online supplemental tables S3 and S4).

### Associations with all-cause and cardiovascular mortality

All adherence and lipid target exposures were associated with all-cause and CVD mortality. For all-cause mortality in unadjusted models, non-achievement of LDL ≤1.8 mmol/L and the non-HDL target was associated with 1.3 times and 2.2 times greater risks, respectively (figure 2A, online supplemental table S5). Following adjustment, this risk remained similar for LDL ≤1.8 mmol/L but attenuated to 1.8 times higher for the non-HDL target. The HRs were nearly identical for CVD mortality, although fewer events resulted in wider CIs (figure 2B, online supplemental table S5).

For average adherence, similar effect sizes were reported for both all-cause and CVD mortality (figure 2, online supplemental tables S6 and S7). In unadjusted analyses, adherence <50% was associated with a hazard of all-cause mortality 1.6 times, and adherence <80% with a hazard 1.5 times, that of those above each threshold. A 10% decrease in adherence was associated with 7% and 6% increases in the risk of all-cause and CVD mortality, respectively, which reduced to 3% in adjusted analyses. For the 50% and 80% thresholds, adjustment attenuated the HRs to 1.2 and 1.4, respectively. In a sensitivity analysis including incomplete years of follow-up in the average adherence calculation, the unadjusted HRs were slightly larger, increasing to two times higher risk for all-cause

mortality, and 1.8 times higher for CVD mortality (online supplemental figure s1, tables S8 and S9). When adjusted, this attenuated to HRs of 1.5 for <50% and 1.7 for <80% adherence for all-cause mortality, and 1.4 and 1.5 for CVD mortality for <50% and <80% adherence, respectively.

### Associations with recurrent MI

In both unadjusted and adjusted models, there were no associations between the achievement of LDL ≤1.8 mmol/L, or statin adherence, and recurrent MIs during follow-up (figure 3, online supplemental tables S10 and S11). Nevertheless, there was an association with the non-HDL target; non-achievement was associated with 21% higher odds of recurrent MI in unadjusted analyses. For all models, adjustment had minimal impact on the ORs. In a sensitivity analysis (online supplemental figure S2 and table S12), where average adherence included incomplete years, these associations remained consistent.

### DISCUSSION

In this Scottish secondary prevention population, 76% achieved LDL ≤1.8 mmol/L, 51% achieved ≥40% reduction in non-HDL, and 85% and 69% had an average statin adherence ≥50% and ≥80%, respectively. Lower average statin adherence, and failure to achieve guideline-recommended lipid targets, were strongly associated with higher risks of all-cause and CVD mortality and were significantly associated with each other. In adjusted models, those with adherence <50% were 24% more likely to die, and those not achieving LDL ≤1.8 mmol/L were 32% more likely to die, with similar patterns observed when cardiovascular causes were considered.

This study advances the literature by the simultaneous comparison of both adherence and lipid targets as surrogates of statin effectiveness, showing these measurements have similar associations with outcomes. However, guidelines do not currently recommend the use of adherence data in estimating the clinical effectiveness of statins in patients. Our data support the notion that a patient's statin adherence, estimated using MPR, may be a useful risk surrogate. Additionally, in this analysis, a 50% threshold appears to have approximately similar associations with outcomes as the, conventionally used, 80% threshold. Given that alternate-day statin dosing is a common approach in routine clinical practice where daily statins are not tolerated,[16] the 50% threshold, therefore, seems pragmatic in this instance. Furthermore, these data also suggest that where adherence data are readily available, annual lipid tests may not be necessary for CVD risk assessment in secondary prevention, potentially reducing costs and workload for health services. However, this hypothesis requires further research.

Lowering LDL has consistently demonstrated significant reductions in mortality within those with established disease.[1] Consistent with this, we report associations of LDL and non-HDL targets with all-cause and CVD specific mortality outcomes. However, while the LDL ≤1.8

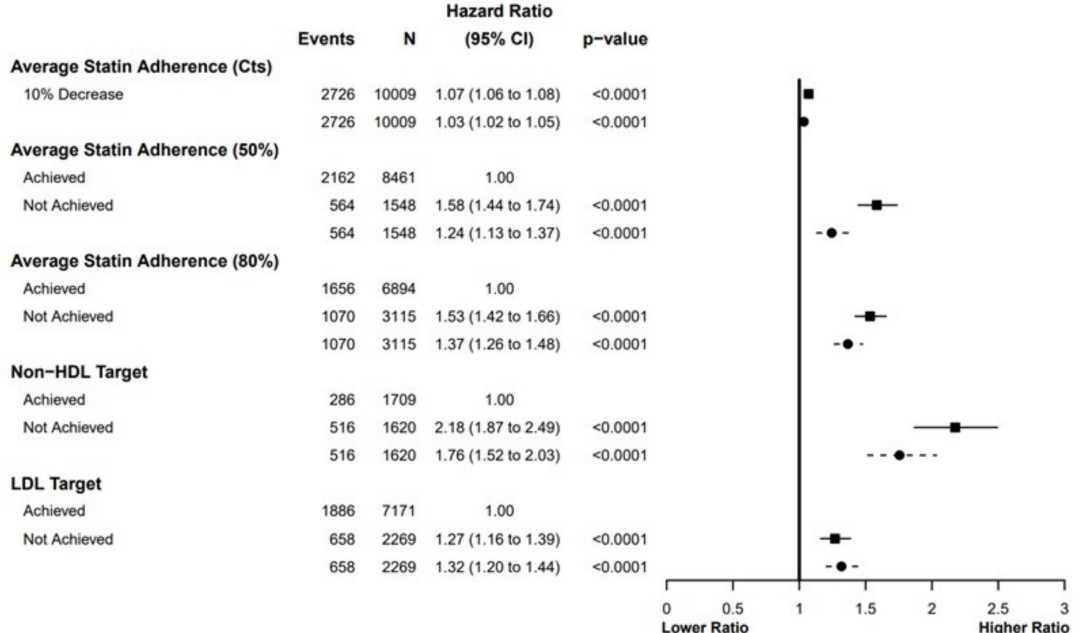

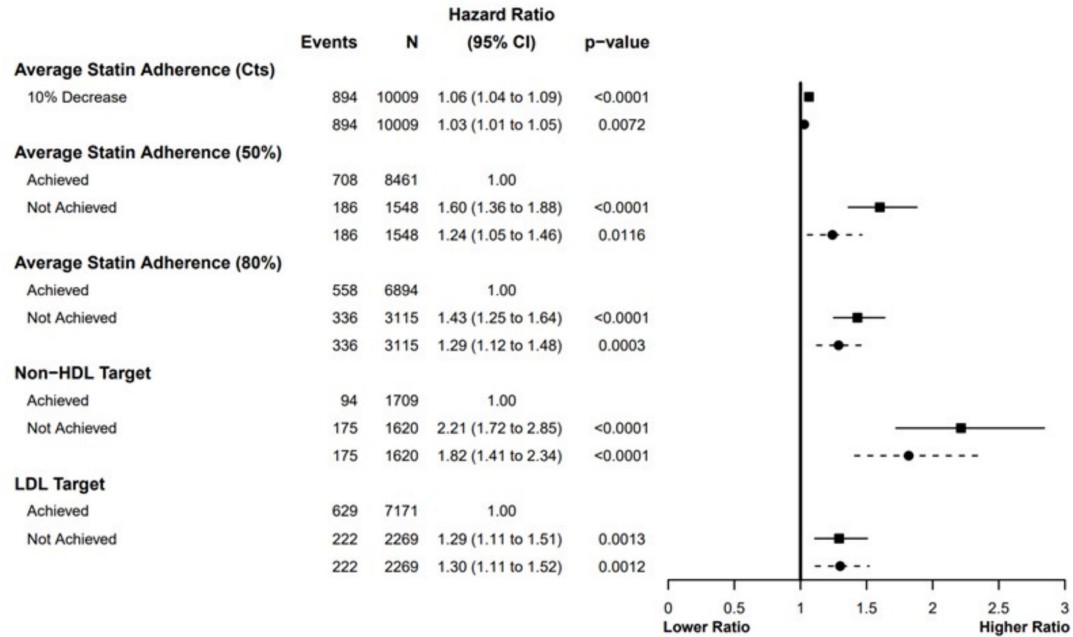

**Figure 2** HRs of (A) all-cause and (B) CVD mortality by average statin adherence and achievement of plasma lipid targets. Ratios presented for unadjusted (solid line and square) and adjusted (dashed line and circle) analyses (adjusted for age, sex, year of MI and SIMD quintile). CVD death defined by ICD10 I00–I99 as the underlying cause. Average adherence calculated as the patient's mean-average annual medication possession ratio of complete years of follow-up. Non-HDL target: ≥40% reduction from a pre-MI baseline, LDL target: ≤1.8 mmol/L. Patients classified as meeting a plasma lipid target if they achieved the target in any year of their follow-up, and not meeting a target otherwise. Cts, continuous; CVD, cardiovascular disease; MI, myocardial infarction; SIMD, Scottish Index of Multiple Deprivation.

mmol/L target used was the one that was recommended by the ESC at the time of data collection, this has since been revised in the latest version to ≤1.4 mmol/L in secondary prevention and ≤1.0 mmol/L after multiple cardiovascular events.[24] This may limit the applicability of these results in future cohorts, although LDL ≤1.8 mmol/L still remains a commonly used target in many

other guidelines.[14] Furthermore, the stronger associations with the non-HDL target than the associations with the LDL target are likely to have arisen due to the nature of the targets, for example, percentage change versus absolute value, rather than the lipid profile components themselves. For example, associations between absolute values of non-HDL and LDL with a composite endpoint

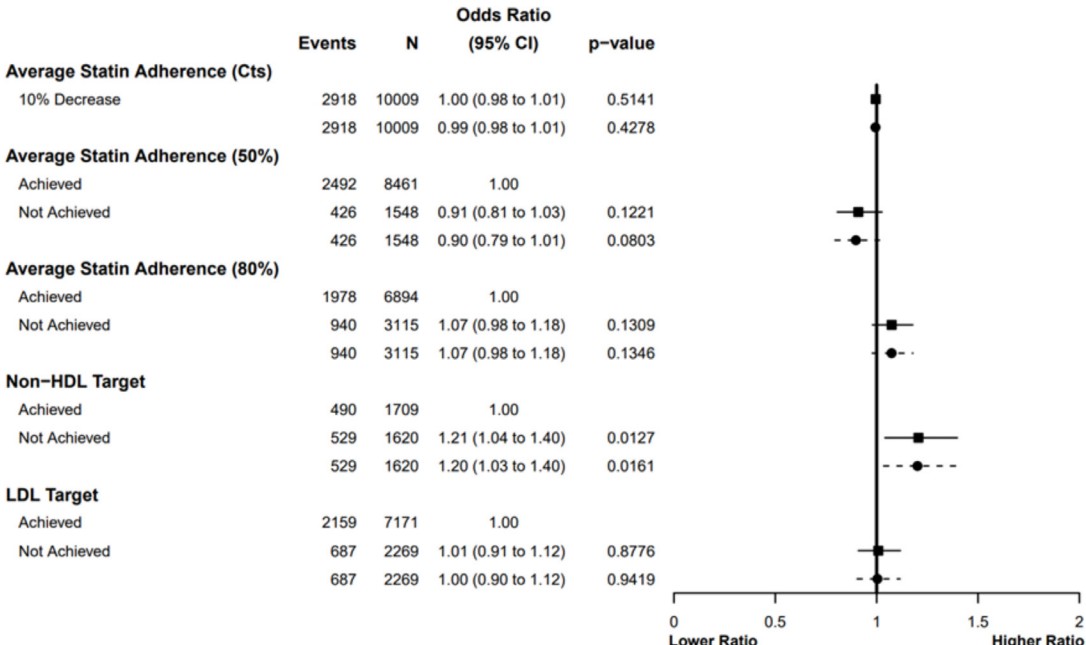

**Figure 3** Odds of MIs during follow-up by average statin adherence and achievement of plasma lipid targets (from NICE and ESC recommendations). Ratios presented for unadjusted (solid line and square) and adjusted (dashed line and circle) analyses (adjusted for age, sex, year of MI and SIMD quintile). CVD death defined by ICD10 I00–I99 as the underlying cause. Models fitted using available case analysis. Average adherence calculated as the patient's mean-average annual medication possession ratio of complete years of follow-up. Non-HDL target: ≥40% reduction from a pre-MI baseline, LDL target: ≤1.8 mmol/L. Patients classified as meeting a plasma lipid target if they achieved the target in any year of their follow-up, and not meeting a target otherwise. Cts, continuous; CVD, cardiovascular disease; ESC, European Society of Cardiology; MI, myocardial infarction; NICE, National Institute for Health and Care Excellence; SIMD, Scottish Index of Multiple Deprivation.

of cardiovascular events are approximately equivalent.[25] Nevertheless, percentage change targets within secondary prevention could prove problematic, with many patients unlikely to suitable baselines for the calculation to be performed. This was evident here where percentage change could only be calculated for 30% of the cohort.

The consistent associations we report between adherence and mortality are also broadly similar to findings observed in other cohorts.[1 7] In high-risk patients in a UK primary care cohort, adherence <80% was associated with a 50% greater hazard of all cardiovascular events following adjustment for demographic factors and comorbidities.[7] Our data suggest that a 50% adherence threshold had a similar association in unadjusted models, although this was attenuated after adjustment. Furthermore, this was also lower than associations observed in a meta-analysis of 12 studies (n=~1 m) who found that patients (in primary and secondary prevention) with poor adherence (<80%) had an 82% greater risk of all-cause mortality. However, although the definition for adherent was harmonised to the 80% threshold to facilitate the meta-analysis, the methods used for determining whether a patient was adherent differed between the included studies, and adjustment for confounders was not consistent.[26]

In this study, there was no association between average statin adherence nor the achievement of lipid targets with recurrent MIs. This second observation is consistent with other observational cohorts where post MI LDL demonstrated poor predictive performance for recurrent atherosclerotic cardiovascular disease events.[27 28] However, statin non-adherence has previously demonstrated a stronger, and intensity-dependent, association with further events in a cohort study in Finland between 2004 and 2016 (n=28 625).[28] One explanation for this could be that in our analysis, average statin adherence and lipid target achievement was determined from information collected before and after recurrent MIs, whereas Lassenius et al used only data for the intervening time.[28] Furthermore, as many patients in our analysis experienced MIs early in follow-up, these exposures are more likely to reflect patient behaviour following multiple MIs, rather than the intervening time. For example, patients may initially increase their adherence or treatment intensity or only achieve lipid targets, after further MIs.[10 28] However, one analysis found only 5% more patients achieved LDL ≤1.8 mmol/L and 7% increased their treatment intensity after their second event.[10]

The associations between statin adherence and lipid target achievement also expand on associations with mortality, with similar associations reported in other observational cohorts using electronic prescribing records. For example, in Georgia, USA (n=~1000), those with adherence <50%, as measured using proportion of days covered (PDC) from dispensing data, were at 88% greater risk of not achieving a 30% reduction in LDL.[11] Similarly, in Greece, another study of patients with established CVD

found that those with adherence <80%, as measured with PDC and following adjustment, were 91% more likely to achieve an LDL ≤1.8 mmol/L after 1 year than those with poor adherence. However, despite this, not all patients who were considered to have good adherence achieved the targeted LDL value from the guidelines.[29] This was also observed in an analysis of UK primary care data using prescribing data, following a cardiovascular event, approximately 70% of patients achieved ≥80% statin adherence (as measured by PDC) in the first year, despite a higher percentage failing to have LDL ≤1.8 mmol/L.[10] Therefore, while adherence appears to be associated with cholesterol levels, it is not the only contributing factor. Likewise, this pattern is observed in this cohort, which calculated MPR, a similar and widely used method, on dispensing data. Nevertheless, a temporal relationship between statin adherence and lipid target achievement cannot be established in these analyses. Both were captured simultaneously during follow-up and, therefore, while an association between the two can be observed, it is difficult to discern, at the population level, its direction.

### Strengths and limitations

This is a contemporary, large dataset from a real-world population. The generalisability of these findings, particularly to the UK population, is therefore good. In this analysis, we confirm and expand on previous research by illustrating that lipid levels and statin adherence are important predictors of death and increase the duration of follow-up beyond the initial few years following statin initiation. This facilitated the estimation of longer-term average statin adherence and more time for target lipid levels to be achieved. Furthermore, the adherence measure used is arguably stronger evidence of medication use due to prescriptions being encashed,[30] rather than prescribed. However, as with all indirect measures of adherence, possession of the medication is not sufficient to ensure patient compliance, and consequently patient adherence may have been overestimated.

In models using average adherence, only those who survived ≥1 year of follow-up were included, reducing the potential for reverse causality. Indeed, in sensitivity analyses, effect sizes were larger for mortality and smaller for lipid target achievement where incomplete time windows were included. However, in all mortality models, time-window bias could have arisen due to differing lengths of follow-up.[31] Specifically, patients who died had reduced opportunities to achieve lipid targets and change their adherence behaviour. The construction of the cohort also meant that patients were assumed to be alive in the absence of a death record, and by allowing for at least 3 years of follow-up to be available (except in the instances of death), this could have resulted in a higher proportion of patients who had moved out of the region. Such patients may be more likely to have higher degrees of frailty or severity of illnesses, and this could have resulted in biases in both the frequency of patient outcomes and their behaviour.

Finally, adjustment for age, sex, deprivation and year of MI, did not substantially alter the effect sizes or their significance. However, this adjustment was not comprehensive, and was limited by the variables contained within the data extract. Indeed, many comorbidities and lifestyle factors which might influence both the exposures and the outcomes in these analyses were not routinely captured. Therefore, other confounders are likely to remain, although the primary aim of this study was to contrast the associations of lipid targets and adherence with outcomes, rather than make causal inferences. For example, adherence in secondary prevention is higher in those with a greater number of comorbidities,[10 11] which may also increase the risk of further events. In this cohort, those experiencing further events were more likely to have a diagnosis of diabetes or had a prior MI; both of which have been associated with higher adherence.[10 12] Consequently, given the observational nature of these data, no causal conclusions are drawn.

### CONCLUSIONS

A significant proportion of the post MI population achieves lipid targets and high average statin adherence, and both have similar strengths of association with reduced mortality. These associations, and those observed between lipid target achievement and adherence, are consistent with the evidence base for statins. We also broadly validate the existing Scottish SIGN non-HDL guidance as clinically relevant, however, low coverage for percentage change variables in secondary prevention limits its practical use in these settings.

**Contributors** JLo and PW conceived and designed the study, and acquired the data. RB conducted statistical analyses and drafted the manuscript. PW checked and validated statistical analyses. JLe and SW contributed to the analytical design and interpretation of the data. JLe, SW, JLo and PW critically revised the manuscript for important intellectual content. RB, JLe, SW, JLo and PW all gave final approval for publication. PW is guarantor and agrees to be accountable for all aspects of work ensuring integrity and accuracy.

**Funding** RB is funded as part of the Medical Research Council Doctoral Training Programme in Precision Medicine (MR/N013166/1).

**Disclaimer** The Medical Research Council has had no role in any aspect of this work.

**Competing interests** RB, JLe, SW and JLo declare no conflict. PW has received grant support from Roche Diagnostics, AstraZeneca, Novartis and Boehringer Ingelheim.

**Patient consent for publication** Not applicable.

**Ethics approval** This study used fully anonymised health record data accessed via a trusted research environment and was therefore exempt from ethical review. Permission was given by NHS GGC Safe Haven local Privacy Advisory Committee (membership includes the ethics committee chair) (project GSH/17/CA/012 PMCVD).

**Provenance and peer review** Not commissioned; externally peer reviewed.

**Data availability statement** Data may be obtained from a third party and are not publicly available. Deidentified data may be obtained by bona fide researchers, for the purposes of research in the public interest, from NHS GGC Safe Haven, via an approval process (https://www.nhsggc.org.uk/about-us/professional-support-sites/glasgow-safe-haven/). Code files are available on request (Rosemary.Brown@glasgow.ac.uk).

**ORCID iDs**
Rosemary Brown http://orcid.org/0000-0002-1719-2761
Jim Lewsey http://orcid.org/0000-0002-3811-8165
Jennifer Logue http://orcid.org/0000-0001-9549-2738

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
