## [Reviewer comments · BMJ Open]

ARTICLE DETAILS

TITLE (PROVISIONAL)	Associations of statin adherence and lipid targets with adverse outcomes in myocardial infarction survivors: a retrospective cohort study
AUTHORS	Brown, Rosemary; Lewsey, Jim; Wild, Sarah; Logue, Jennifer; Welsh, Paul

VERSION 1 – REVIEW

REVIEWER	Giannakoulas, George Aristotle University of Thessaloniki, Cardiology Department
REVIEW RETURNED	27-Jul-2021

GENERAL COMMENTS	Brown et al performed this retrospective cohort study to examine associations between statin adherence and lipid target achievement in myocardial infarction (MI) survivors, and their associations with mortality and recurrent MIs. They found that over 4.5 years follow-up, 76% achieved LDL \leq1.8mmol/L, and 84.5% had average adherence \geq50%. In univariable models, not meeting LDL\leq1.8mmol/L was associated with increased risks of all-cause mortality (HR 1.27 (1.16, 1.39), $p<0.0001$) and CVD mortality (1.29 (1.11, 1.51), $p=0.0013$). Adherence $<$50% was associated with increased risks of all-cause mortality (1.58 (1.44, 1.74), $p<0.0001$) and CVD mortality (1.60 (1.36, 1.88), $p<0.0001$). Adjustment for confounders did not abrogate these associations. Neither exposure was associated with recurrent MIs. This study although not novel in design and findings reinforces the importance of achievement of lipid and adherence targets in secondary prevention, especially in patients post-MI. The paper is well written, findings are clearly presented and I think that the overall quality of the manuscript is good, with great clinical relevance. Some comments Why did authors chose the 50% of adherence as the primary exposure of interest? The message that 50% and 80% of adherence provide similar clinical outcome results should be downgraded (also in Discussion, page 11. Line 3-7), since this is not supported by other studies. Please discuss the findings of this study and compare with the Chowdhury et al EHJ 2013 meta-analysis paper. Moreover, a major limitation of these type of studies is the use of the national prescription data base to infer compliance and adherence to the prescribed drugs which therefore does not
---

	equate to actual compliance. This should be highlighted in the Limitations section. Another limitation is that authors used the 2016 European Society of Cardiology (ESC) target for LDL cholesterol ≤ 1.8mmol/l for secondary prevention which nowadays after the publication of the new 2019 European Society of Cardiology Guidelines with the recommendation of lower LDL targets may seem obsolete. Please comment on Limitation section. In the Results section, authors should provide the mean value of adherence of the total population. Authors included patients who experienced a non-fatal MI hospital admission between January 2009 and July 2014, followed up from the date of MI admission until July 2017 or death. They should provide more up-to-date information and compare their findings with more contemporary data such as those of Zafeiropoulos et al, Atherosclerosis 2021. They found that patients with good adherence had greater mean LDL-C change from baseline [32.8 ± 40.4 mg/dl vs. 19.8 ± 38.1 mg/dl; $p=0.02$], lower LDL-C values [median 69.0 mg/dl (IQR, 55.5–88.5) vs. 80.0 mg/dl (64.0–109.0); $p=0.004$] than those with poor adherence and were more likely to achieve the LDL-C goal (adjusted odds ratio 1.91, 95% CI 1.09 to 3.41; $p=0.02$). Minor comments: Page 3, line 7: “and their associations with recurrent MIs and mortality”. Please change to “and their associations with mortality and recurrent MIs.” Page 10, line 5: “(Figure 2, Tables s6-7), with.” Modify sentence. Page 10, line 10: “all-cause and circulatory mortality respectively,”. Modify to “all-cause and CVD mortality respectively”. Page 11, line 15-17: “Whilst on-statin lipid targets are somewhat controversial [24,25], and there is a lack of robust evidence to support their use in guidelines [14],”. Please modify sentence since the aim of current paper is not to review the on-statin lipid targets in secondary prevention.
--	---

REVIEWER	Yang, Zhirong University of Cambridge, Primary Care Unit
REVIEW RETURNED	05-Aug-2021

GENERAL COMMENTS	There have been some studies on this topic. The findings were similar. Could the authors please provide further justification for this study in the introduction section? Could the authors also discuss those studies in comparisons with this study? To investigate the potential impact of statin adherence on the achievement of the lipid targets, the lipid levels should be measured after the time point of defining statin adherence. It seems unclear what the time window was used to define the achievement of the lipid targets. The temporal relationship
---

	between statin adherence and the achievement of the lipid targets should be made clearer. The authors used logistic regression for recurrent MIs but used Cox regression for mortality. Why were different models used? In the modelling, possible confounders in the adjustment only covered age at MI, sex, deprivation quintile, and year of MI. In fact, many other factors, e.g., multimorbidity, co-medication and lifestyle, have not been controlled in this analysis. I would suggest further adjustment for these factors. To allow for at least three years follow-up, patients who did not have an MI between 1st January 2009 and 31st July 2014 were excluded. This exclusion was based on the status of follow-up instead of baseline characteristics and therefore may lead to selection bias. Besides, it were unclear why there should be at least three years follow-up. According to the flowchat, 56170 patients were excluded because the postcode was not within NHS GGC. This exclusion criterion was not mentioned in the text. Also, why were people of this kind excluded? The modelling results on the associations between the adjusted confounders and the outcomes showed in most tables were not of interest in this study. Perhaps they could be removed? Potential information bias in using routine data for epidemiological research should have been discussed.
--	---

VERSION 1 – AUTHOR RESPONSE

Reviewer: 1

Dr. George Giannakoulas, Aristotle University of Thessaloniki

1. Brown et al performed this retrospective cohort study to examine associations between statin adherence and lipid target achievement in myocardial infarction (MI) survivors, and their associations with mortality and recurrent MIs. They found that over 4.5 years follow-up, 76% achieved LDL \leq 1.8mmol/L, and 84.5% had average adherence \geq 50%. In univariable models, not meeting LDL \leq 1.8mmol/L was associated with increased risks of all-cause mortality (HR 1.27 (1.16, 1.39), $p < 0.0001$) and CVD mortality (1.29 (1.11, 1.51), $p = 0.0013$). Adherence $<$ 50% was associated with increased risks of all-cause mortality (1.58 (1.44, 1.74), $p < 0.0001$) and CVD mortality (1.60 (1.36, 1.88), $p < 0.0001$). Adjustment for confounders did not abrogate these associations. Neither exposure was associated with recurrent MIs.

This study although not novel in design and findings reinforces the importance of achievement of lipid and adherence targets in secondary prevention, especially in patients post-MI. The paper is well written, findings are clearly presented and I think that the overall quality of the manuscript is good, with great clinical relevance.

Response: Thank you for this summary and comments regarding the quality of this manuscript.

2. Why did authors chose the 50% of adherence as the primary exposure of interest? The message that 50% and 80% of adherence provide similar clinical outcome results should be downgraded (also in

Discussion, page 11. Line 3-7), since this is not supported by other studies. Please discuss the findings of this study and compare with the Chowdhury et al EHJ 2013 meta-analysis paper.

Response: Thank you for your comment. The 50% adherence was chosen as the primary exposure as alternate day dosing is often used in clinical settings in patients where statins are poorly tolerated. Within the data available, it would not be known whether such a regime was recommended to these patients. Therefore, the 50% adherence choice seemed pragmatic within the circumstances. This has now been added to the methods section:

“Given that alternate day dosing is often implemented instances of poor tolerance of medication [16], the 50% threshold was considered the primary exposure of interest.”

The messages regarding similarity have also been downgraded as follows in the abstract and discussion:

- Abstract: “Conclusions Non-achievement of lipid and adherence targets are associated with increased risks of all-cause and CVD mortality.”
- Discussion: “Additionally, in this analysis, a 50% threshold appears to have approximately similar associations with outcomes as the, conventionally used, 80% threshold. Given that alternate-day statin dosing is a common approach in routine clinical practice where daily statins are not tolerated [16], the 50% threshold, therefore, seems pragmatic in this instance.”

The Chowdhury paper is also discussed in the discussion as follows:

- “Our data suggest that a 50% adherence threshold had a similar association in unadjusted models, although this was attenuated after adjustment. Furthermore, this was also lower than associations observed in a meta-analysis of 12 studies (n=~1m) who found that patients (in primary and secondary prevention) with poor adherence (<80%) had an 82% greater risk of all-cause mortality. However, although the definition for adherent was harmonised to the 80% threshold to facilitate the meta-analysis, the methods used for determining whether a patient was adherent differed between the included studies, and adjustment for confounders was not consistent [26].”

3. Moreover, a major limitation of these type of studies is the use of the national prescription data base to infer compliance and adherence to the prescribed drugs which therefore does not equate to actual compliance. This should be highlighted in the Limitations section.

Response: Thank you for highlighting this. This has now been added to the limitations section as detailed below:

- “Furthermore, the adherence measure used is arguably stronger evidence of medication use due to prescriptions being encashed [29], rather than prescribed. However, as with all indirect measures of adherence, possession of the medication is not sufficient to ensure patient compliance, and consequently patient adherence may have been overestimated.”
4. Another limitation is that authors used the 2016 European Society of Cardiology (ESC) target for LDL cholesterol $\leq 1.8\text{mmol/l}$ for secondary prevention which nowadays after the publication of the new 2019 European Society of Cardiology Guidelines with the recommendation of lower LDL targets may seem obsolete. Please comment on Limitation section.

Response: Thank you for highlighting this. The 2016 target was used as this was the one recommended during data collection and is still used in many other international guidelines. This has been highlighted within the discussion, with the new targets outlined:

- “Consistent with this, we report associations of LDL and non-HDL targets with all-cause and CVD specific mortality outcomes. However, whilst the $LDL \leq 1.8$ mmol/l target used was the one that was recommended by the ESC at the time of data collection, this has since been revised in the latest version to ≤ 1.4 mmol/l in secondary prevention and ≤ 1.0 mmol/l after multiple cardiovascular events [24]. This may limit the applicability of these results in future cohorts, although $LDL \leq 1.8$ mmol/l still remains a commonly used target in many other guidelines [14]. Furthermore, the stronger associations with the non-HDL target...”

5. In the Results section, authors should provide the mean value of adherence of the total population.

Response: Thank you for highlighting this omission. This has now been added to the results section:

- “Over an average 4.5 years follow-up, the mean-average statin adherence was 79.6%, and 84.5% of patients achieved $\geq 50\%$ average statin adherence and 68.9% had $\geq 80\%$.”
6. Authors included patients who experienced a non-fatal MI hospital admission between January 2009 and July 2014, followed up from the date of MI admission until July 2017 or death. They should provide more up-to-date information and compare their findings with more contemporary data such as those of Zafeiropoulos et al, *Atherosclerosis* 2021. They found that patients with good adherence had greater mean LDL-C change from baseline [32.8 ± 40.4 mg/dl vs. 19.8 ± 38.1 mg/dl; $p=0.02$], lower LDL-C values [median 69.0 mg/dl (IQR, 55.5–88.5) vs. 80.0 mg/dl (64.0–109.0); $p=0.004$] than those with poor adherence and were more likely to achieve the LDL-C goal (adjusted odds ratio 1.91, 95% CI 1.09 to 3.41; $p=0.02$).

Response: Thank you for highlighting this. The dates included and the length of follow-up available were determined by the timing of the data extract, and therefore we do not have access to more up-to-date information at this time. The study you reference has now been discussed within the discussion.

- “The associations between statin adherence and lipid target achievement also expand on associations with mortality, with similar associations reported in other observational cohorts using electronic prescribing records. For example, in Georgia, USA ($n \sim 1,000$), those with adherence $< 50\%$, as measured using proportion of days covered (PDC) from dispensing data, were at 88% greater risk of not achieving a 30% reduction in LDL [11]. Similarly, in Greece, another study of patients with established CVD found that those with adherence $< 80\%$, as measured with PDC and following adjustment, were 91% more likely to achieve an $LDL \leq 1.8$ mmol/l after one year than those with poor adherence. However, despite this, not all patients who were considered to have good adherence achieved the targeted LDL value from the guidelines [29]. This was also observed in an analysis of UK primary care data utilising prescribing data, following a cardiovascular event, approximately 70% of patients achieved $\geq 80\%$ statin adherence (as measured by PDC) in the first year, despite a higher percentage failing to have $LDL \leq 1.8$ mmol/l [10]. Therefore, whilst adherence appears to be associated with cholesterol levels, it is not the only contributing factor.”

7. Page 3, line 7: “and their associations with recurrent MIs and mortality”. Please change to “and their associations with mortality and recurrent MIs.”

Response: Thank you for this comment. This has now been amended in the abstract, and introduction to read:

- Abstract: “To examine associations between statin adherence and lipid target achievement in myocardial infarction (MI) survivors, and their associations with mortality and recurrent MIs.”
- Introduction: “This analysis, using routinely collected data available for a secondary prevention population representative of a UK health board region, seeks to investigate associations between statin adherence and lipid target achievement, and their associations with mortality and recurrent MIs.”

8. Page 10, line 5: “(Figure 2, Tables s6-7), with.” Modify sentence.

Response: Thank you for highlighting this. The sentence has now been modified to read:

- “For average adherence, similar effect sizes were reported for both all-cause and CVD mortality (Figure 2, Tables s6-7).”

9. Page 10, line 10: “all-cause and circulatory mortality respectively,”. Modify to “all-cause and CVD mortality respectively”.

Response: Thank you for highlighting this. This has now been amended:

- “A 10% decrease in adherence was associated with 7% and 6% increases in the risk of all-cause and CVD mortality respectively, which reduced to 3% in adjusted analyses.”

10. Page 11, line 15-17: “Whilst on-statin lipid targets are somewhat controversial [24,25], and there is a lack of robust evidence to support their use in guidelines [14],”. Please modify sentence since the aim of current paper is not to review the on-statin lipid targets in secondary prevention.

Response: Thank you for your comment. The first sentence of this paragraph has now been amended to remove this implication.

- “Lowering LDL has consistently demonstrated significant reductions in mortality within those with established disease [1].”

Reviewer: 2

Dr. Zhirong Yang, University of Cambridge

11. There have been some studies on this topic. The findings were similar. Could the authors please provide further justification for this study in the introduction section? Could the authors also discuss those studies in comparisons with this study?

Response: Thank you. We agree that previous studies have looked at similar topics, but the contrast between adherence and lipid targets as proxy surrogates of CVD risk here is important. We have updated the introduction to state

- Therefore, there superficially appears to be differences between lipid-lowering medication adherence and lipid target achievement. This leads to uncertainties in clinical guidelines and in clinical practice as to whether adherence or lipid targets are more useful surrogates of risk reduction [14].

This analysis, using routinely collected data available for a secondary prevention population representative of a UK health board region, therefore seeks to compare and contrast statin

adherence and lipid target achievement, and compare their associations with mortality and recurrent MIs. This analysis also investigates which is more strongly associated with outcomes, and therefore relevant for clinical decision making.

We have updated the discussion to state

- The consistent associations we report between adherence and mortality are also broadly similar to findings observed in other cohorts [1,7]. In high-risk patients in a UK primary care cohort, adherence <80% was associated with a 50% greater hazard of all cardiovascular events following adjustment for demographic factors and comorbidities [7]. Our data suggest that a 50% adherence threshold had a similar association in unadjusted models, although this was attenuated after adjustment. Furthermore, this was also lower than associations observed in a meta-analysis of 12 studies (n~1m) who found that patients (in primary and secondary prevention) with poor adherence (<80%) had an 82% greater risk of all-cause mortality. However, although the definition for adherent was harmonised to the 80% threshold to facilitate the meta-analysis, the methods used for determining whether a patient was adherent differed between the included studies, and adjustment for confounders was not consistent [26].
- The associations between statin adherence and lipid target achievement also expand on associations with mortality, with similar associations reported in other observational cohorts using electronic prescribing records. For example, in Georgia, USA (n~1,000), those with adherence <50%, as measured using proportion of days covered (PDC) from dispensing data, were at 88% greater risk of not achieving a 30% reduction in LDL [11]. Similarly, in Greece, another study of patients with established CVD found that those with adherence <80%, as measured with PDC and following adjustment, were 91% more likely to achieve an LDL≤1.8mmol/l after one year than those with poor adherence. However, despite this, not all patients who were considered to have good adherence achieved the targeted LDL value from the guidelines [29]. This was also observed in an analysis of UK primary care data utilising prescribing data, following a cardiovascular event, approximately 70% of patients achieved ≥80% statin adherence (as measured by PDC) in the first year, despite a higher percentage failing to have LDL≤1.8mmol/l [10]. Therefore, whilst adherence appears to be associated with cholesterol levels, it is not the only contributing factor. Likewise, this pattern is observed in this cohort, which calculated MPR, a similar and widely used method, on dispensing data. Nevertheless, a temporal relationship between statin adherence and lipid target achievement cannot be established, in these analyses. Both were captured simultaneously during follow-up and therefore, whilst an association between the two can be observed, it is difficult to discern, at the population level, its direction.

12. To investigate the potential impact of statin adherence on the achievement of the lipid targets, the lipid levels should be measured after the time point of defining statin adherence. It seems unclear what the time window was used to define the achievement of the lipid targets. The temporal relationship between statin adherence and the achievement of the lipid targets should be made clearer.

Response: Thank you for highlighting this. Adherence is the average measure over the course of follow up, and the lipid target could be obtained at any point during the follow up. It is therefore acknowledged that the temporal relationship between them cannot be established in this analysis. This has been added to the discussion when discussing these models as a limitation of their results:

- “Nevertheless, a temporal relationship between statin adherence and lipid target achievement cannot be established, in these analyses. Both were captured simultaneously during follow-up and therefore, whilst an association between the two can be observed, it is difficult to discern, at the population level, its direction.”

13. The authors used logistic regression for recurrent MIs but used Cox regression for mortality. Why were different models used?

Response: We now clarify

- “Due to the potential for multiple recurrent MIs to occur in an individual, and the relatively short one-year time windows modelled, logistic regression was used to model the associations between average adherence and the achievement of lipid targets, and the association between adherence, and lipid targets, with recurrent MIs.”

14. In the modelling, possible confounders in the adjustment only covered age at MI, sex, deprivation quintile, and year of MI. In fact, many other factors, e.g., multimorbidity, co-medication and lifestyle, have not been controlled in this analysis. I would suggest further adjustment for these factors.

Response: Thank you for highlighting this. Our perspective here was that of a decision-making clinician; we aimed to explore associations as informational tools, rather than causal mechanisms. In addition, adjustment was limited by the data that was available in the data extract. For example, primary care records were not available, and therefore many morbidities and lifestyle factors were not able to be accurately derived. This has now been clarified within the limitations section.

- “However, this adjustment was not comprehensive, and was limited by the variables contained within the data extract. Indeed, many comorbidities and lifestyle factors which might influence both the exposures and the outcomes in these analyses were not routinely captured. Therefore, other confounders are likely to remain, although the primary aim of this study was to contrast the associations of lipid targets and adherence with outcomes, rather than make causal inferences.”

15. To allow for at least three years follow-up, patients who did not have an MI between 1st January 2009 and 31st July 2014 were excluded. This exclusion was based on the status of follow-up instead of baseline characteristics and therefore may lead to selection bias. Besides, it were unclear why there should be at least three years follow-up.

Response: Thank you for the comment. The cohort was constructed for those with at least three years follow up as the focus was on the longer term follow up of these patients rather than the time around the initial event. However, it is important to note that patients could have a shorter period of time in the cohort if death occurred in the first three years (these patients were not excluded if their MI was between the dates stated). To improve clarity, this has now been clarified within the methods.

- “MIs were identified in linked hospital admission records (ICD10: I21, I22 in any diagnosis field) to identify patients who experienced an MI between 1st January 2009 and 31st July 2014. These patients were followed up from their first non-fatal MI admission until 31st July 2017 or death, whichever occurred first. Patients who only experienced fatal MIs (death occurred during admission or within 30 days of discharge), whose statin adherence was >200%, or were missing a socioeconomic deprivation index were removed.”

Selection bias arising from the extended period of follow up is now highlighted as a limitation in the discussion as follows:

- “The construction of the cohort also meant that patients were assumed to be alive in the absence of a death record, and by allowing for at least three years of follow-up to be available (except in the instances of death), this could have resulted in a higher proportion of patients who had moved out of the region. Such patients may be more likely to have higher degrees of frailty or severity of illnesses, and this could have resulted in biases in both the frequency of patient outcomes and their behaviour.”

16. According to the flowchart, 56170 patients were excluded because the postcode was not within NHS GGC. This exclusion criterion was not mentioned in the text. Also, why were people of this kind excluded?

Response: Thank you for this comment. Patients whose postcodes were from outside NHS GGC were excluded as these patients were likely transferred into the area for a hospital admission only and therefore it was unlikely that the data would contain complete prescribing or lipid testing records. Their exclusion was mentioned briefly in the text, but this has now been updated in the methods to include justification:

- “We excluded patients who resided outside the region, as these individuals are likely to have travelled into the region for only some aspects of their care, and therefore a complete representation of their health status was unlikely. Patients who were <18 years, died before 1st January 2009, had multiple death certificates, or received apparent posthumous lipid tests or prescriptions for lipid-lowering medications more than six months posthumously were also excluded.”

17. The modelling results on the associations between the adjusted confounders and the outcomes showed in most tables were not of interest in this study. Perhaps they could be removed?

Response: Thank you for this suggestion. The association with the adjusted confounders were not included within the main manuscript. However, they were included as part of the summary tables within the supplementary material for completeness purposes and therefore have not been removed.

18. Potential information bias in using routine data for epidemiological research should have been discussed.

Response: We agree there is potential for some misclassification of exposures and outcomes, although this is generally less of a concern here than with self-reported data or interviewer data. We now say

- However, as with all indirect measures of adherence, possession of the medication is not sufficient to ensure patient compliance, and consequently patient adherence may have been overestimated.
- The construction of the cohort also meant that patients were assumed to be alive in the absence of a death record, and by allowing for at least three years of follow-up to be available (except in the instances of death), this could have resulted in a higher proportion of patients who had moved out of the region. Such patients may be more likely to have higher degrees of frailty or severity of illnesses, and this could have resulted in biases in both the frequency of patient outcomes and their behaviour.

VERSION 2 – REVIEW

REVIEWER	Giannakoulas, George Aristotle University of Thessaloniki, Cardiology Department
REVIEW RETURNED	04-Sep-2021
GENERAL COMMENTS	All edits performed and improved manuscript quality